# Formation of Structure and Properties of Two-Phase Ti-6Al-4V Alloy during Cold Metal Transfer Additive Deposition with Interpass Forging

**DOI:** 10.3390/ma14164415

**Published:** 2021-08-06

**Authors:** Yuri Shchitsyn, Maksim Kartashev, Ekaterina Krivonosova, Tatyana Olshanskaya, Dmitriy Trushnikov

**Affiliations:** Welding Manufacturing, Technology of Materials & Metrology Department, Perm National Research Polytechnic University, Komsomolsky Prospekt, 29, 614990 Perm, Russia; schicin@pstu.ru (Y.S.); goncharsk@mail.ru (M.K.); katerinakkkkk@mail.ru (E.K.); tvo66@mail.ru (T.O.)

**Keywords:** Ti-6Al-4V, titanium alloys, additive manufacturing, CMT deposition, structure, properties

## Abstract

The paper deals with the main formation patterns of structure and properties of a titanium alloy of the Ti-6Al-4V system during additive manufacturing using cold metal transfer (CMT) wire deposition. The work aims to find the optimal conditions for layer-by-layer deposition, which provides the high physical and mechanical properties of the titanium alloy of the Ti-6Al-4V system hybrid, additively manufactured using CMT deposition. Particular attention is paid to interpass forging during the layered printing of the product. Additionally, we investigate how the heat treatment affects the structure and properties of the Ti-6Al-4V alloy that has been CMT-deposited, both with and without forging. These studies have shown that the hybrid multilayer arc deposition technology, with interpass strain hardening, allows the use of high temperature and high technology titanium alloys to obtain products of a required geometric shape. It has been proven that the interpass deformation effect during CMT deposition contributes to a significant decrease in the sizes of the primary β-grains. In addition, forging enhances the effect of microstructure refinement, which is associated with phase recrystallization in deformed areas. It is shown that the heat treatment leads not only to a change in the morphology of the phases but also to additional phase formations in the structure of the Ti-6Al-4V-deposited metal while the mechanism is realized and consists of the gradual decomposition of the martensitic α′-phase and the formation of a dispersive α_2_-phase. This structure formation process is accompanied by the dispersion hardening of the α-phase. The strength characteristics of the Ti-6Al-4V alloy obtained using layer-by-layer CMT with forging are given; they exceed the strength level of materials obtained with the traditional technologies of pressure treatment, and there is no decrease in plasticity characteristics. The use of the subsequent heat treatment makes it possible to increase the ductility characteristics of the deposited and forged Ti-6Al-4V material by 1.5–2 times without strength loss.

## 1. Introduction

Additive arc processes, which make it possible to synthesize large metal parts with a minimum allowance for machining and properties at the level of metals of traditional technologies, ensure significant savings of metal and reduce the costs of pre-production engineering [1,2,3,4]. This concerns the production of aluminum [5,6], titanium [7,8,9,10], and other light high-strength alloys.

Among the arc methods, CMT (cold metal transfer) is an updated version of consumable electrode welding in a shielded gas based on the mechanism of a controlled metal flow mode into the weld pool using pulsed current and reciprocating wire movement. It has a number of great advantages when it comes to additive technologies [9]. The wire reverse feed system is synchronized with a high-speed digital control that determines the arc length, short circuit phase, and heat transfer to the weld zone [9]. This process excludes metal splashing and ensures the stability of weld formation and minimal heat transfer to the processing area. The CMT process can be successfully applied to aerospace materials such as high-strength and heat-resistant titanium alloys.

However, there are some structural features of titanium alloys that have significant disadvantages for welding and deposition technologies, in particular, an unfavorable dendritic structure, anisotropy, banding due to the transcristallite adjustment of dendritic clusters, and, thus, a decrease in mechanical properties (ultimate tensile strength is below 850 MPa), i.e., loss of plasticity [7,10,11].

The layer-by-layer deformation effect is a solution to these problems, which can increase the density of the deposited material and refine the structure, thus obtaining higher mechanical properties. Improvement of the microstructure and strength characteristics of the material occurs during the deformation on each layer of the deposited solid material (3–5 mm thick) by rolling, forging, or ultrasonic shock treatment [12,13,14,15,16]. The subgrain microstructure formed in this way does not disappear during the fusion of subsequent layers, i.e., it turns out to be thermally stable. The positive effect of the deformational processing by rolling on the strength of products during layer-by-layer synthesis is described in [13]. The use of interpass forging, in addition to reducing technological residual stresses and changing their sign on the surface of the product from tensile to compressive ones, leads to a decrease in grain sizes, an increase in strength characteristics to the level of the base metal, and a decrease in porosity and anisotropy [14].

The use of ultrasonic shock treatment on welded joints made of titanium alloy provides the formation of surface compressive residual stresses in them and an increase in the fatigue characteristics of the weld material by more than 6 times [15]. The layer-by-layer deformation effect is capable of increasing the density of the deposited material, refining the structure and, thus, obtaining higher mechanical properties during the additive plasma deposition of aluminum alloys [16].

The purpose of this work is to determine the optimal conditions for layer-by-layer deposition and provide the high mechanical properties of the titanium alloy of the Ti-6Al-4V system, obtained with the additive manufacturing of workpieces in hybrid additive manufacturing using CMT. Particular attention is paid to the interpass impact deformation effect by forging during the layer-by-layer printing of the product as well as the effect of the heat treatment on the structure and properties of the CMT-produced Ti-6Al-4V alloy with and without forging.

## 2. Technologies, Investigation Techniques, and Materials

The Ti-6Al-4V deposition material, which is based on the Ti-6Al-4V ternary system characteristic of most high-temperature titanium alloys, belongs to two-phase (α + β) alloys. According to the principles of the heat-resistant alloying of titanium alloys, the multi-component alloy is alloyed with α-stabilizers (Al), β-stabilizers (V, Fe), and neutral elements (O, C, N), which provides an effective combination of dispersion and solid solution hardening mechanisms. The chemical composition of the deposition wire is given in Table 1.

Figure 1 shows a schematic diagram of the wire-arc three-dimensional CMT consumable electrode deposition. The deposition was carried out using a 3-axis CNC milling machine of the Hybrid Additive Manufacturing Group of Companies (Perm, Russia) and a Fronius TransPuls Synergic 5000 CMT power supply of Fronius (Wels, Austria). To protect the deposited metal from ambient air, a protective device with an argon supply was used. Forging was carried out using a SA7401H AIRPRO pneumatic hammer of Airpro Industry Corp. (New Taipei City, Taiwan). All this equipment was installed in the AT-300 machining center of the Hybrid Additive Manufacturing Group of Companies for the implementation of hybrid additive manufacturing technology. The AT-300 machining center is shown in Figure 2. The heat treatment was carried out in a vacuum furnace; the mode included temperature 850 °C, holding time for 2 h, and cooling with a furnace.

Vertical walls with square section specimens with interpass strain hardening were deposited according to a pre-selected deposition mode: direct polarity, arc current I = 140–152 A; arc voltage U = 14–16 V; wire feed speed V_wf_ = 6.0 m/min; deposition speed V_d_ = 60 cm/min; arc length correction (ALC) ALC = −30%; dynamic correction (DC) (influences of the current waveform and duration of the CMT cycle) DC = −5%; volume of argon supplied to the welding torch Q_torch_ = 7.5 L/min; volume of argon supplied to the protective device Q_pd_ = 80 L/min.

The optimal strategy of filling was used for the deposition of a multilayer specimen by the method of a staggered order of layers, with the distance between adjacent beads in one layer of 8 mm and the height of one layer of 2.5 mm. The filling strategy is designed in such a way that specimens for tensile tests and metallographic analysis can be made from the deposited specimen.

Interpass forging was applied after the deposition of each individual bead. Forging was carried out according to a pre-selected mode: the speed of the hammer movement V = 100 mm/min (the linear number of blows N/L = 28.2 blows/mm at the standard frequency of hammer blows N = 2820 blows/min); bits tip is the hemisphere with radius R = 20 mm; impact energy E = 19.74 J; sample temperature T = 250 °C—temperature to which the deposited bead cooled down before applying forging. The pressing force of the pneumatic hammer was 300 N, and it was generated by a pneumatic cylinder. The relative linear deformation of the layer due to the use of forging is 6% since the height of the unforged layer is 1.86 mm and the height of the forged layer is 1.75 mm.

To study the mechanical and macro- and microstructural characteristics of the deposited metal, we deposited workpieces in the form of a flat wall (Figure 3) from which test specimens were cut. To identify the effect of forging on the anisotropy of properties, the specimens were prepared for tensile tests in two directions: in the horizontal X-direction, along the deposition direction, and in the vertical Z ones.

The microstructure studies used an Altami CM0745-T stereomicroscope, an Altami MET 1T inverted light microscope with a magnification of up to 1000 times, and Altami Studio 3.5 software. Phase composition studies used the results of X-ray phase analysis.

The microhardness was measured using a PMT-3 hardness tester of LLC “MTPK-LOMO” (Saint-Petersburg, Russia) at a load of 200 g, 200 measurements on each sample. The measurements were carried out in the form of two parallel tracks of 100 measurements with a step of 0.15 mm, as shown in Figure 4. The results were statistically processed.

Tensile tests were performed using an Instron 8801 servo-hydraulic of Instron Corp. (Norwood, MA, USA) testing system focused on dynamic and static testing.

## 3. Investigations of the Macrostructure of the Deposited Layers of the Ti-6Al-4V Alloy during CMT

Figure 5 shows fragments of the macrostructure of the deposited metal after deposition with and without forging.

On the macrosections of the metal obtained by CMT deposition (CMT deposition is obtained on Fronius TransPuls Synergic 5000 CMT power supply) in the optimal mode without forging, it can be seen that a coarse-grained macrostructure is formed, and large columnar grains formed during the growth of primary β-grains through 3–4 layers of the deposition and have clear boundaries with a thin transition zone (Figure 5a).

Interpass forging during CMT deposition leads to a significant decrease in the size of the primary β-grains; the size of these grains is limited to 1–2 deposition beads. The boundaries between primary β-grains are also clearly visible (Figure 5b).

Further heat treatment does not lead to a significant change in the size of the primary β-grains, both during deposition without forging and with it (see Figure 6). However, in this case, the boundaries between the primary β-grains are not clearly revealed, which obviously indicates an intense diffusion process along the grain boundaries during high-temperature holding under heat treatment. For the same reason, the boundaries between the deposited layers in the macrostructure of the heat-treated specimens are weakly manifested (Figure 6a), and they are not revealed at all on the specimens deposited using forging (Figure 6b).

## 4. Microstructure Analysis of the Deposited Layers of the Ti-6Al-4V Alloy during CMT Deposition

A typical structure of the titanium alloy of the Ti-Al-V system is observed: a small amount of the initial β-grain (a body-centered cubic β-phase; in fact, it is guessed only from thin interlayers along the grain boundaries), a lamellar α-phase (hexagonal close-packed α-phase), and coarsely acicular α′-martensite. For a more detailed analysis of the microstructure of the deposited wall, the following sections were investigated at different magnifications, i.e., the boundary between the primary β-grains and the boundary between the deposited layers and the intragranular structure.

### 4.1. Microstructure of Deposited Layers of Ti-6Al-4V Alloy without Forging

The intragranular microstructure of the deposited layers is a mixture of the lamellar α-phase, coarsely acicular α′-martensite, and the β-metastable phase (Figure 7 and Figure 8). Along the boundaries of most of the primary β-grains, precipitates of the α-phase (α′-phase), with a thickness of about 1–2 μm, are observed (Figure 7). The boundary between the deposited layers is most clearly visible at low magnification (×100) due to various etching amounts of the layers (Figure 8). At high magnifications, the formation of the α-phase and its partial coagulation can be observed.

In general, the microstructure is characterized by the presence of certain orientations of the structural components, which are realized during the martensitic transformation.

### 4.2. Microstructure of Deposited Layers of Ti-6Al-4V Alloy with Forging

The phase composition of the intragranular regions during the deposition with forging does not change: the microstructure contains α′-martensite, α and β-metastable phases (Figure 9 and Figure 10); however, the forging significantly changes the morphology of these phases: a decrease in the size of primary β-grains (from 500 to 100 μm, on average) and an increase in the dispersion of the formed α- and α′-phases. The length of the α′-martensite needles and the dimensions of the α-phase plates decrease by 15–20%. At the boundaries of some primary β-grains, the α-phase is also precipitated, while the thickness of the α′-phases decreases (Figure 9). At some boundaries between the deposited layers, slight coagulation of the α-phase is observed (Figure 10).

The observed changes in the morphology of the phases during the deposition with forging can be explained, taking into account the known principles of structural transformations during pressure treatment of titanium alloys from [17,18]:-In the case of deformation at temperatures in the α + β region, both the β-phase and α(α′)-phase take part in the deformation process;-The lower the temperature of deformation, the greater the amount of deformable α(α′)-phase, while the refinement of the shape of inter-boundary α(α′)-plates and near-border α-trim is observed;-As a result of the deformation in the β-phase, the density of dislocations increases, and in the α(α′)-phase, the density of deformation twins and the degree of subgrain structure increase.

In our case, the surface deformation of the deposited layer was carried out at a temperature below 300 °C, below the temperatures of martensitic transformation and recrystallization. When a subsequent layer is applied, the deformed layer experiences heating in the temperature range above the temperatures of phase transformation and recrystallization but below the melting point. In the area heated above the temperature of the phase transformation α + β → β, the existing deformed β-phase undergoes recrystallization processes. The deformed α(α′)-phase comes to the β-phase with the formation of new grains in places with an increased degree of defectiveness, i.e., twins and subgrain structure. As a result, a new structure is formed in this area, consisting of a large number of small β-grains of an equiaxed shape. With the subsequent cooling, taking into account the small volume of this area and high cooling rates, the β-phase undergoes a reverse transformation with the formation of α′-martensite, α, and β-metastable phases within each new grain with the formation of a more dispersed intragranular structure.

When the deformed section is heated to a temperature range below the phase transformation temperature and above the recrystallization temperature, recrystallization processes occur in both β and α(α′) phases, which lead to the refinement of only the intragranular structure within the former primary β-grain. This is due to the fact that the recrystallization of the α-phase is accompanied by the retention of the shape and position of deformed α-plates and α-colonies [17].

Thus, the refinement of the macrostructure during CMT deposition with layer-by-layer deformation is closely related to the processes of phase recrystallization in the deformed areas.

These conclusions are confirmed by the results of scanning electron microscopy taken at different magnifications, shown in Figure 11 for the CMT-deposited metal using layer-by-layer deformation (at the bottom) and without it (at the top). During deposition with layer-by-layer deformation, a structure is formed with α(α′)-phase plates of a smaller thickness and an obvious and pronounced acicular shape.

The results of X-ray phase analysis, presented in Figure 12, show that the structure of the deposited metal contains an α-phase, a martensite α(α′)-phase, and a β-phase. The difficulty in identifying phases in titanium alloys of this type is that the martensite phase α′ has the same hexagonal close-packed lattice as the α-phase, with similar crystal lattice parameters; thus, the X-ray diffraction of the α′ and α phases is practically indistinguishable. The α′ phase differs only by a large smearing of the interference maxima [15]. The peaks in the diffraction patterns are characteristic for the following substances:-Ti hexagonal (space group P63/mmc; No. 96-151-2548 in the database) α(α′)-phase;-Ti cubic (space group Im-3m; No. 96-900-8555 in the database) β-phase.

## 5. Analysis of the Effect of Heat Treatment on the Microstructure of the Deposited Layers of the Ti-6Al-4V Alloy during CMT Deposition

The performed studies have shown that heat treatment (HT) leads not only to a change in the morphology of the phases but to additional phase formations in the structure of the Ti-6Al-4V-deposited metal.

In both deposition options, with forging and without it (Figure 13 and Figure 14, respectively), the intragranular microstructure acquires a basket-like structure, the length of the α-phase plates decreases, the width increases, and the plates have a so-called branch-like look, sometimes divided into a chain of separate α-particles. At the boundaries between the primary β-grains with the precipitation of the α(α′)-phase, an increase in the width of this α-trim is observed (left fragments in Figure 13 and Figure 14). On the specimens of the CMT deposition with forging, the boundaries between the deposited layers are not visible (the right fragment of Figure 11), and on the specimens with the CMT deposition (without forging), an increase in the size of α-phase particles is observed between some of the deposited layers; their coagulation and partial globulation occur (the right fragment of Figure 10).

Our studies have shown that the heat treatment leads not only to a change in the morphology of the phases but to additional phase formations in the structure of the Ti-6Al-4V-deposited metal.

During the heat treatment of the deposited Ti-6Al-4V alloy, the mechanism of structure formation is most likely realized, which consists of the gradual decomposition of the martensitic α′-phase and the formation of the dispersion α_2_-phase.

At the first stage of annealing, at a temperature of 800 °C, the decomposition of α′-martensite occurs. The process begins with a redistribution of the alloying elements and the formation of regions in the α′-phase with an equilibrium content of the alloying elements. In these areas, the separation of the α-phase occurs. As a result of this, α′-martensite is additionally saturated with β-stabilizers: α′→α′_saturated_ + α. The process ends with the partial separation of a small amount of the β-phase from the α′_saturated_ martensite.

After cooling to a temperature of 600 °C at the second stage of annealing, the process of ordering occurs in the martensitic phase α′_saturated_, leading to the formation of coherent particles with a completely ordered structure, separated by an inhomogeneous α-phase: α′(α″)_saturated_ → α + α_2_. The α_2_-phase is based on the Ti_3_Al compound. The final microstructure after the two-stage annealing is a mixture of phases: α + α_2_ + β. This hardening is called the dispersion hardening of the α-phase [17,18].

In the images of the microstructure of the thermally treated specimens obtained with the scanning electron microscope, we can see the “branch-like” nature of the plates and their division into individual particles. At the same time, on such individual particles, precipitates of the dispersed α_2_-phase in the form of white sections are observed (Figure 15 and Figure 16).

The phase composition of the deposited metal formed after the heat treatment is confirmed by the results of the X-ray phase analysis shown in Figure 17. Phase α_2_, which is based on the Ti_3_Al compound, has a crystal lattice close to that of the α-phase. It differs in that aluminum atoms occupy not random but quite definite places in the lattice (lattice parameters of the α_2_-phase: a = 2a_α_, c = c_α_). The X-ray diffraction patterns of the α_2_-phase usually differ from those of the α-phase by the presence of extra superstructural reflexes [17]. The peaks in the diffraction patterns are characteristic for the following substances:-Ti hexagonal (space group P63/mmc; No. 96-151-2548 in the database) α(α′)-phase;-Ti cubic (space group Im-3 m; No. 96-900-8555 in the database) β-phase;-Ti 3Al hexagonal (space group P63/mmc; No. 96-153-2768 in the database) α_2_-phase.

## 6. Testing the Mechanical Properties of the Ti-6Al-4V-Deposited Alloy Synthesized with the CMT Technology

Table 2 presents the results of measurements of the microhardness of the specimens under study and the statistical analysis of the obtained results. Figure 18 shows the histograms of frequency distributions of microhardness values Vickers hardness (HV) 0.2. The analysis of the results shows that the layer-by-layer deformation during deposition contributes to an increase in the average value of microhardness, which is associated with a decrease in the dimension of the formed structural units as well as an increase in structure fineness during interlayer forging during deposition. Heat treatment, which leads to the dispersion hardening of the deposited material, additionally increases the hardness of both depositions, with forging and without it.

The results of testing the strength and plastic properties of the deposited material are shown in Table 3. A table with representative tensile data can be found in the Appendix A (Table A1) at the end of the article.

To identify the effect of forging and heat treatment on the anisotropy of properties, the specimens were prepared for tensile tests in two directions: in the horizontal X-direction, along the deposition direction, and in the vertical Z ones. Overall, there were 4 samples for each direction for every type of deposition. Table 3 shows the average value and absolute error, with reliability equal to 0.95. A schematic diagram of the method of cutting out specimens for mechanical testing and structural analysis is shown in Figure 19.

The test results show that the anisotropy of properties in the vertical and horizontal directions is negligible. The deposited material has sufficiently high mechanical properties. Subsequent heat treatment of unforged specimens does not lead to a significant increase in the mechanical properties of the deposited metal. The use of interpass forging during electric arc deposition makes it possible to improve the structure of the deposited metal and increase its strength (1050–1060 MPa). Subsequent heat treatment of forged specimens leads to additional refinement of macro- and microstructures and makes it possible to increase ductility (relative elongation up to 13–14%) at a high level of strength of the deposited metal (980–1000 MPa). The samples made by three-dimensional deposition with interpass deformation treatment and subsequent heat treatment show high quality, characterized by an almost complete absence of anisotropy and a guaranteed level of mechanical properties for the material of forgings made of Ti-6Al-4V alloy.

## 7. Conclusions

The influence of interpass deformation treatment and subsequent heat treatment on the quality of the deposited metal during wire-arc three-dimensional deposition was investigated. It has been found that layer-by-layer deformation during CMT deposition contributes to a significant decrease in the size of primary β-grains, i.e., the size of these grains is limited by 1–2 deposition beads, in contrast to 3–4 beads for deposition without forging. In addition, forging enhances the effect of microstructure refinement, which is associated with phase recrystallization in deformed areas: a structure is formed with plates of α(α′)-phases of a smaller thickness and an obvious acicular shape.It was found that heat treatment leads not only to a change in the morphology of the phases but to additional phase formations in the structure of the Ti-6Al-4V-deposited metal; in this case, a mechanism is realized, which consists of the gradual decomposition of the martensitic α′-phase and the formation of a dispersion α_2_-phase. The final microstructure after the two-stage annealing is a combination of phases: α + α_2_ + β. This process of structure formation is accompanied by the dispersion hardening of the α-phase.It has been revealed that the strength characteristics of the Ti-6Al-4V alloy obtained using layer-by-layer CMT deposition with forging exceed the strength level of the materials obtained with the traditional technologies of pressure treatment, while no decrease in plasticity characteristics occurred in this case. The use of the subsequent heat treatment makes it possible to increase the plasticity characteristics of the deposited and forged Ti-6Al-4V material by 1.5–2 times without loss of strength.

## Figures and Tables

**Figure 1 materials-14-04415-f001:**
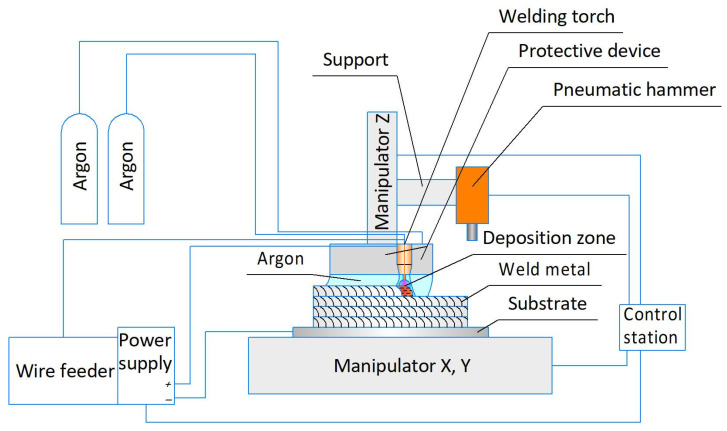
Diagram of the wire-arc three-dimensional consumable electrode deposition.

**Figure 2 materials-14-04415-f002:**
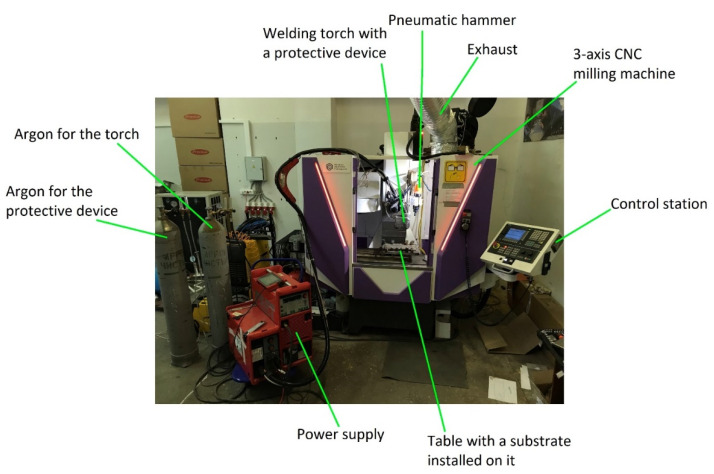
AT-300 machining center of the Hybrid Additive Manufacturing Group of Companies for the implementation of hybrid additive manufacturing technology.

**Figure 3 materials-14-04415-f003:**
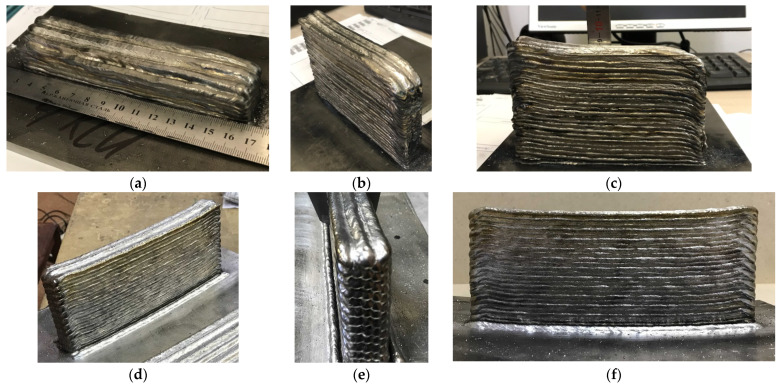
Specimens obtained by multilayer deposition: (**a**) an example of a deposited specimen for cutting horizontal specimens for mechanical tests; (**b**–**f**) an example of a deposited specimen for cutting vertical and horizontal specimens for mechanical tests.

**Figure 4 materials-14-04415-f004:**
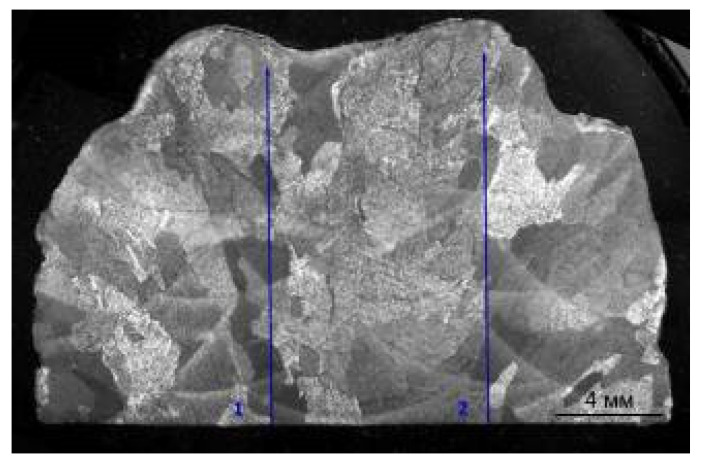
An example of the microhardness measurements of the deposited samples. The measurements were carried out in the form of two parallel tracks of 100 measurements with a step of 0.15 mm.

**Figure 5 materials-14-04415-f005:**
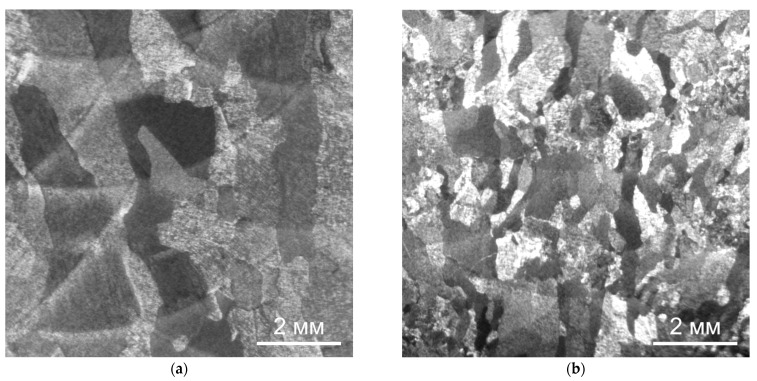
Macrostructure of the Ti-6Al-4V alloy during CMT deposition: (**a**) without forging, (**b**) with interpass forging.

**Figure 6 materials-14-04415-f006:**
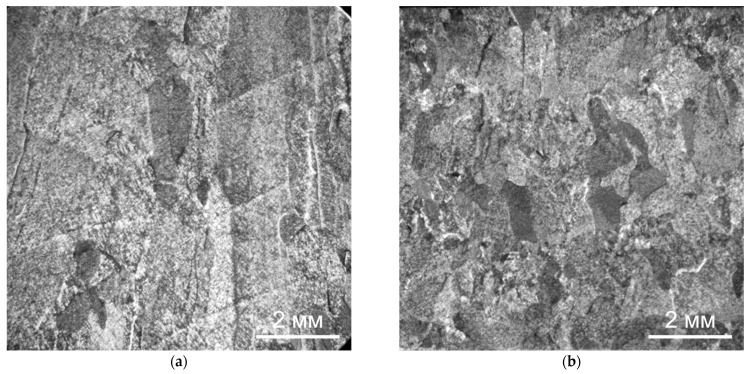
Macrostructure of the Ti-6Al-4V alloy during CMT deposition with heat treatment: (**a**) without forging, (**b**) with interpass forging.

**Figure 7 materials-14-04415-f007:**
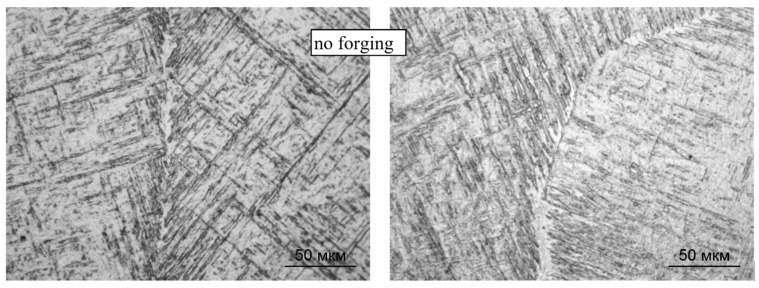
Boundaries between primary β-grains in the metal structure during CMT deposition without forging; ×500.

**Figure 8 materials-14-04415-f008:**
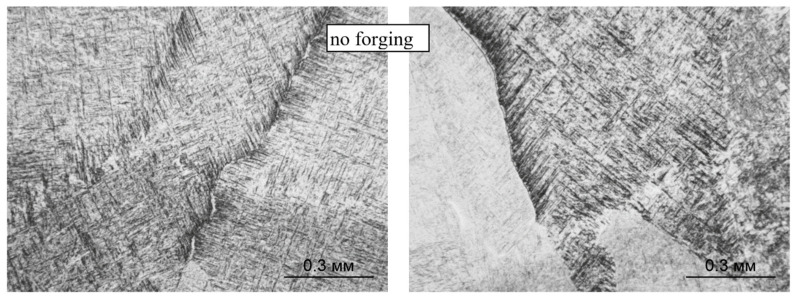
Boundaries between the deposited layers in the structure of the metal CMT deposition without forging; ×100.

**Figure 9 materials-14-04415-f009:**
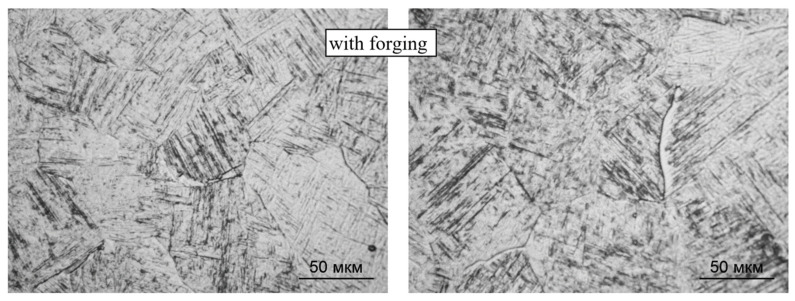
Boundaries between primary β-grains of CMT deposition with forging; ×500.

**Figure 10 materials-14-04415-f010:**
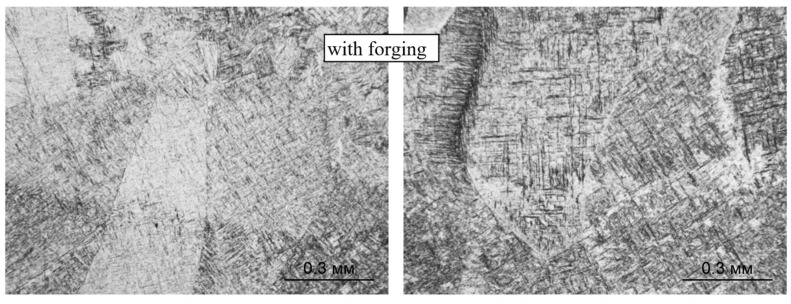
Boundaries between the deposited layers during CMT deposition with forging; ×100.

**Figure 11 materials-14-04415-f011:**
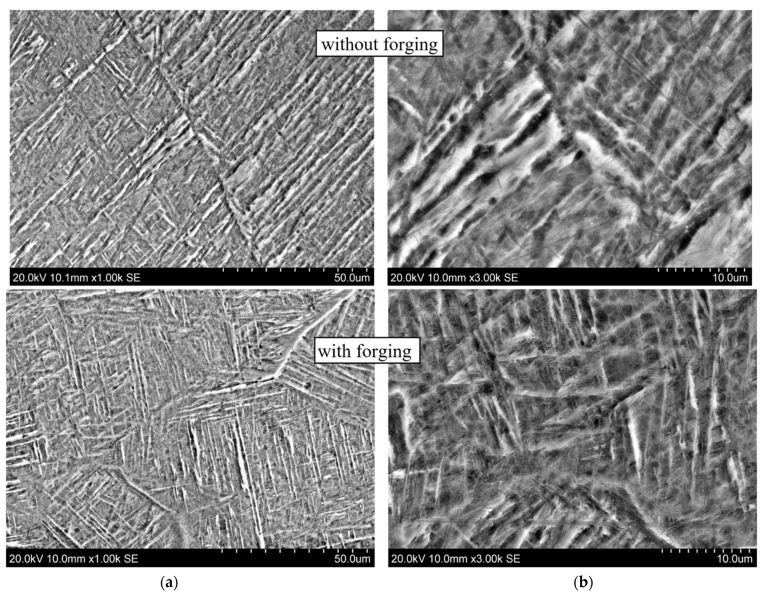
Microstructure of alloy Ti-6Al-4V. CMT deposition without and with forging at different magnifications: (**a**) ×1000; (**b**) ×3000.

**Figure 12 materials-14-04415-f012:**
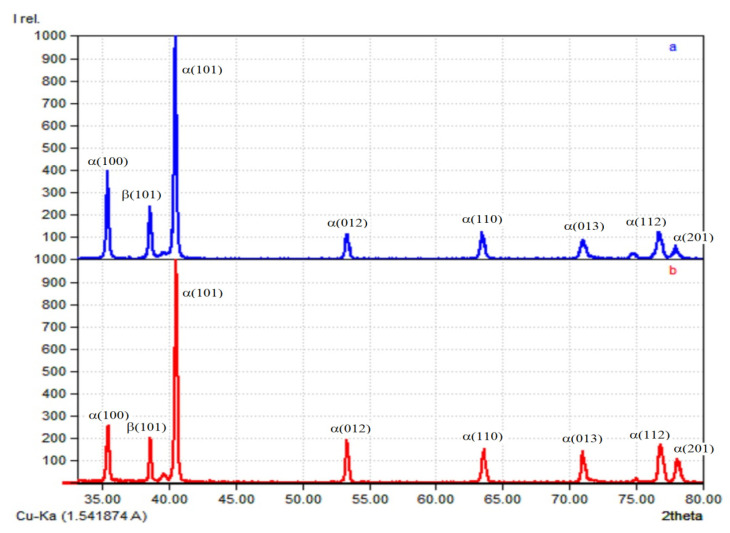
Diffractogram of specimens obtained by CMT deposition: (**a**) without forging; (**b**) with forging.

**Figure 13 materials-14-04415-f013:**
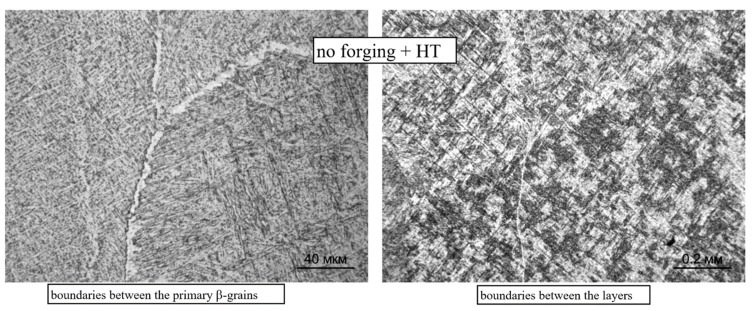
Microstructure and boundaries between primary β-grains (×500) and between layers (×100) during CMT deposition without forging after two-stage annealing.

**Figure 14 materials-14-04415-f014:**
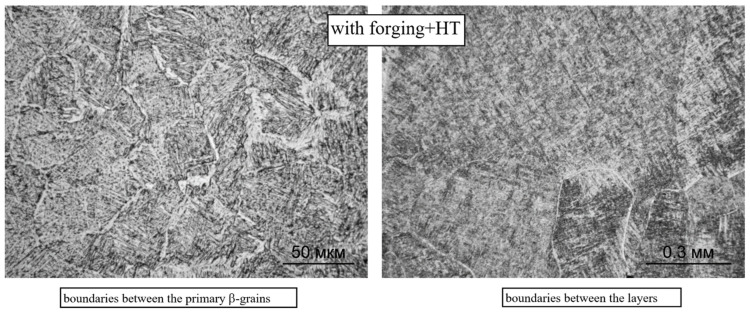
Microstructure and boundaries between primary β-grains (×500) and between layers (×100) during CMT deposition with forging after two-stage annealing (×500).

**Figure 15 materials-14-04415-f015:**
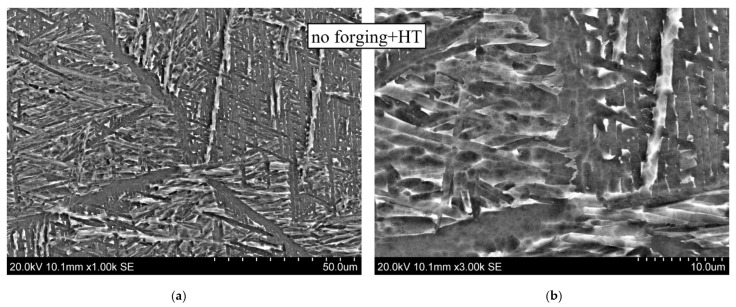
Microstructure of CMT deposition after two-stage annealing: (**a**) ×1000; (**b**) ×3000.

**Figure 16 materials-14-04415-f016:**
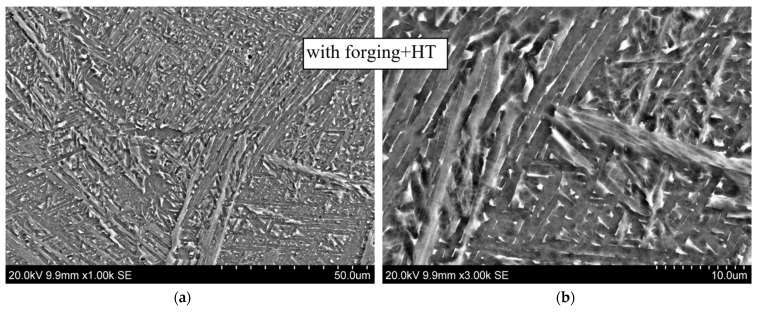
Microstructure of CMT deposition with forging after two-stage annealing: (**a**) ×1000; (**b**) ×3000; (**c**) ×10,000.

**Figure 17 materials-14-04415-f017:**
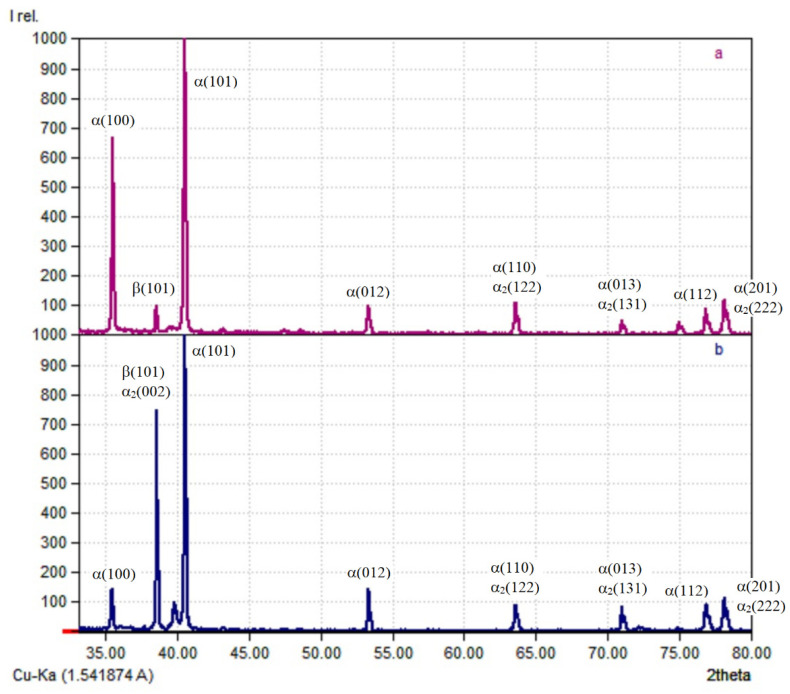
Diffraction patterns for specimens obtained using CMT deposition (**a**) and CMT deposition with forging (**b**) after heat treatment.

**Figure 18 materials-14-04415-f018:**
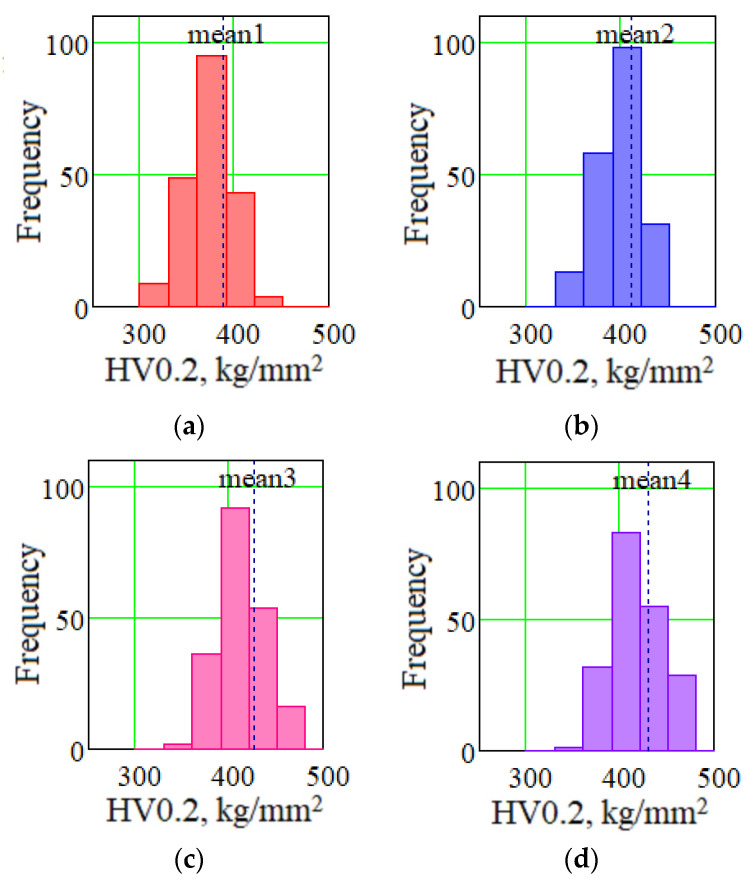
Block diagrams of frequency distributions of microhardness values HV 0.2 kg/mm^2^: (**a**) CMT deposition; (**b**) CMT deposition with forging; (**c**) CMT deposition with heat treatment; (**d**) CMT deposition with forging and heat treatment.

**Figure 19 materials-14-04415-f019:**
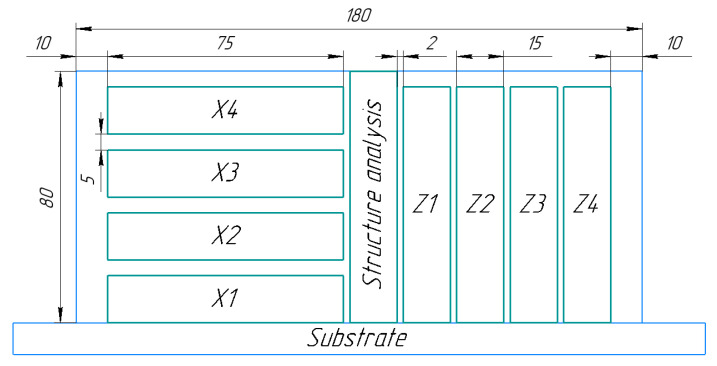
A schematic diagram of the method of cutting out specimens for mechanical testing and structural analysis: X1–X4 specimens for tensile tests in the horizontal direction, along the deposition direction; Z1–Z4 specimens for tensile test in the vertical direction.

**Table 1 materials-14-04415-t001:** Chemical composition of Ti-6Al-4V alloy according to AMS 4928.

Alloy Grade	Ti	Al	V	Fe	C	O	N	H	Other Elements
Ti-6Al-4V	Base	5.5–6.75	3.5–4.5	<0.3	<0.08	<0.20	<0.05	<0.0125	<0.40

**Table 2 materials-14-04415-t002:** Microhardness of specimens obtained using different options, HV 0.2 kg/mm^2^.

Type of Deposition	Number of Tested Specimens	Average Values	Minimum Values	Maximum Values	RMSD
CMT deposition without forging and without heat treatment	2	388.7	320	447	23.6
CMT deposition without forging and with heat treatment	2	428.3	367.5	484.3	25.5
CMT deposition with forging and without heat treatment	2	412	360.2	463	23.5
CMT deposition with forging and with heat treatment	2	430.7	371	477.6	24.5

**Table 3 materials-14-04415-t003:** Deposition modes and mechanical properties of specimens: ultimate tensile strengths (UTS), yield strengths (YS), and elongation (Elong.).

Type of Deposition	Number of Tested Specimens	UTS (MPa)	YS (MPa)	Elong. (%)
AMS 4999 (Ti-6Al-4V, additive manufacturing) min, X/Z	-	899/855	800/765	6/5
AMS 4928 (Ti-6Al-4V, forging after annealing), X/Z	-	896/-	827/-	10/-
CMT deposition without forging and without heat treatment, X/Z	X 4/Z 4	950 ± 40/930 ± 20	850 ± 50/ 850 ± 20	7 ± 4/6 ± 3
CMT deposition without forging and with heat treatment, X/Z	X 4/Z 4	995 ± 12/970 ± 14	920 ± 20/880 ± 20	7.7 ± 0.4/6.1 ± 1.4
CMT deposition with forging and without heat treatment, X/Z	X 4/Z 0	1056 ± 8/-	970 ± 13/-	8 ± 3/-
CMT deposition with forging and with heat treatment, X/Z	X 4/Z 4	995 ± 16/1000 ± 8	905 ± 50/920 ± 13	13 ± 1/13 ± 3
Ti-6Al-4V, AMS 4999 (LMD)	-	855–889	765–800	at least 5–6

## Data Availability

Data sharing is not applicable for this article.

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
