# Peer review of "Formation of Structure and Properties of Two-Phase Ti-6Al-4V Alloy during Cold Metal Transfer Additive Deposition with Interpass Forging"

_materials, 2021, doi:10.3390/ma14164415_

Round 1
Reviewer 1 Report
In this article, Shchitsyn et al. have reported the structure formation of Ti-6Al-4V alloy and its properties during additive manufacturing through Cold Metal Transfer (CMT) deposition under different conditions. The authors have well-structured their manuscript and the results were discussed. The reviewer has some comments that must be integrated as follows.
- In my opinion, citation of references should not be started by “the authors of [XY]”. Please paraphrase the facts/findings and add reference number at the end. Or, alternatively, if you want to mention authors, also introduce their names instead.
- In table 1 and 3, the authors could consider using points (.) instead of comma (,) for reported numbers – to align with the text.
- In table 2, under “type of deposition”, the conditions of forging and heat treatment were not clearly mentioned for each specimen. Please align with table 3.
- In table 2 and 3, authors should point out, how many specimens per condition were tested.
- In table 3, on specimen Nr. 5, I think there is a mistake: it should be “Deposition with forging without heat treatment, XY/Z”.
- In table 3, to what was subscription “B” and “T” in strength referring to? Please clarify. Additionally, the authors should show measured values as average together with standard deviation.
- For the mechanical tests (Figure 3), the authors mentioned to have measured the sample in the horizontal XY. The reviewer guesses, the authors measured them along the printed direction. Did the authors consider also to measure the mechanical properties in perpendicular to the printed direction? This would give valuable information for the adhesion between the printed layers in the same plain.
Reviewer 2 Report
This paper describes a fabrication method using cold metal transfer and a forging process after each deposited layer. The paper is not clearly presented. The presentation of this paper should be improved. Some questions and comments are listed below.
- CMT in the title should be spelled out. This would be clearer to the readers.
- Abstract, lines 16-18. The sentence reads very strange. The authors should rephrase it.
- beta-grains was used starting from line 20. Do authors mean β-stabilizers (V, Fe,) grain? The symbol should be defined before usage. Or do the authors actually mean the beta-phase of titanium alloy. The authors should make it clear.
- ??-grains. Strange symbol shows starting from line 142. The symbol should be defined before usage.
- alpha- alpha'-phases were used starting from line 186. Does α-phase relate to α-stabilizers (Al)? This is confusing. The authors should define symbols before usage. Or do authors actually mean a hexagonal close-packed α phase and a body centered cubic β phase for titanium alloys?
- page 7, line 188, alpha-phase is also precipitated. A brief description about how authors judge that is alpha precipitates will help readers.
- What is the forging force and the deformation of the sample after forging?
- SigmaB sigmaT and tau should be defined before usage.
- The representative tensile test data should be shown for readers reference. At least it should be shown in the supplementary information.
Round 2
Reviewer 1 Report
The authors have revised the manuscript based on reviewers’ comments and suggestions. However, two points from the reviewer should still be considered and adjusted:
- On behalf of the microhardness test, it was not clear the authors only did 2 measurements per sample or several measurements within 2 tested specimens. The reviewer also found that the procedure on how to measure the microhardness is missing. The authors should describe this under sector “2. Technologies, Investigation Techniques and Materials”
- The authors have already added a table with tensile data in the appendix. So, the authors should consider reporting the results in Table 3 in the form of average values ± standard deviation, instead of reporting minimum and maximum values, which are already visible in the appendix.
After the rework on these two points, the reviewer believes that the manuscript can be published in Materials.
Reviewer 2 Report
The authors have responded to the reviewers' comments.
Author Response
Thank you for your consideration of this manuscript.